Adult picky eating and associations with childhood picky eating, maternal feeding, aversive sensory responsiveness, disgust and obsessive-compulsive symptoms

Zohar Ada H. 1 2 adaz@ruppin.ac.il
Barhum Shapira Dorin 1
http://orcid.org/0000-0001-5565-2506 Lev-Ari Lilac 1 2
Bachner-Melman Rachel 1 3
1 Clinical Psychology Graduate Program, Ruppin Academic Center , Emek Hefer , Israel
2 The Lior Zfaty Center for Suicide and Mental Pain Studies, Ruppin Academic Center , Emek Hefer , Israel
3 Paul Baerwald School of Social Work and Social Welfare, Hebrew University Jerusalem , Jerusalem , Israel
Cimino Silvia
Electronic publication date: 2025 May 16
Publication date: 2025
Volume: 13
Electronic Location ID: e19444
Received 2024 Nov 25; Accepted 2025 Apr 17
Copyright: © 2025 Zohar et al.
Copyright year: 2025
Copyright holder: Zohar et al.
License: This is an open access article distributed under the terms of the Creative Commons Attribution License, which permits unrestricted use, distribution, reproduction and adaptation in any medium and for any purpose provided that it is properly attributed. For attribution, the original author(s), title, publication source (PeerJ) and either DOI or URL of the article must be cited.
License URL: https://creativecommons.org/licenses/by/4.0/

Keywords: Picky eating, Aversive sensory responsiveness, Disgust, Obsessive-compulsive disorder, Parental feeding practices

Funding: The authors received no funding for this work.

==============================
Background

Adult picky eating (PE) can cause distress, malfunction, and malnutrition. The purpose of this study was to examine adult PE, its relationship to childhood PE and to symptoms of obsessive-compulsive disorder (OCD), sensory processing disorder, food and general disgust, and maternal child feeding practices.

Methods

Adult participants (N = 772; 636 women) self-reported on measures of adult PE (Adult Picky Eating Questionnaire (APEQ)), obsessive-compulsive symptoms (Obsessive-Compulsive Inventory–Revised (OCI-R)), sensory processing difficulties (Sensory Responsiveness Questionnaire–Intensity Scale (SRQ-IS)), general disgust propensity (Disgust Propensity and Sensitivity Scale-12 (DPSS-12)), food disgust (Food Disgust Scale–short (FDS-S)), and three maternal feeding practices (Retrospective Child Feeding Questionnaire (RCFQ)).

Results

Childhood PE and current adult PE were strongly associated. Adults with PE who scored in the top 25 percentile on the APEQ were at risk for OCD and sensory processing disorder; most reported having been picky eaters in childhood. Structural equation modelling with good fit indices confirmed a developmental model in which general disgust, food disgust and sensory processing difficulties contributed to childhood PE and maternal feeding practices (pressure to eat, restriction), which in turn contributed to OCD symptoms. In addition, pressure to eat, restriction and OCD symptoms contributed directly to adult PE.

Conclusions

Severe adult PE is related to childhood PE and disgust sensitiviy, OCD and sensory processing disorder. Adults with severe PE may meet criteria for adult avoidant restrictive feeding or eating disorders. For adults with extreme PE clinical intervention may be warranted.

Introduction

Even though picky eating (PE) is not defined as a disorder, in many young adults it is far from benign. In a survey of adult community volunteers, around half of those with PE also reported disordered eating (Wildes, Zucker & Marcus, 2012). In college students, PE was associated with symptoms of stress, anxiety, depression, obsessive-compulsive disorder (OCD), social eating anxiety, inflexible eating, disordered eating, dietary restraint and binge-eating (Barnhart et al., 2021). Fox et al. (2023) found that PE in young adults was associated with the perception of social and physical threats, which mediated the negative correlation between PE and eating-related quality of life. Whereas some degree of food selectivity is normative, adults who report consuming less than ten food items, reject entire food groups, are rigid about food presentation, reject food as bitter or sour, or disengage from meals, are at risk for disordered eating and low quality of life (Fox et al., 2023). It is therefore important to screen for PE. In this study, we aim to identify a cut-off score of clinical significance using the Adult Picky Eating Questionnaire (APEQ; Ellis et al., 2017).

PE during childhood is common and often transient (Zohar et al., 2020; Zohar, 2022). However, it may have long-term effects. In a cohort followed from birth, most 23-year-old adults with PE reported that they were children with PE, and the longer the duration of childhood PE, the greater the likelihood of PE during early adulthood (Van Tine et al., 2017). Pereboom et al. (2023) assessed children’s PE via parental report at age four, and self-reported eating preferences at age 18. Childhood PE predicted the consumption of a restricted range of vegetables, fish and dairy products at age 18.

Parents who perceive their children as having PE may respond in a variety of ways. They may feel distressed and guilty and thus turn the process of feeding and eating into an emotionally fraught process (Steinsbekk et al., 2017; Zohar, Lev-Ari & Bachner-Melman, 2019), increasing their child’s food fussiness and PE. They may pressure the child to eat, cajoling or coercing, which typically backfires (Ellis et al., 2016; Galloway et al., 2006). Children with PE who do not eat the nutritious food they are offered, may make up missing calories by eating high calorie snack foods that parents then try to limit. However, there is no evidence that doing so improves PE outcome. In a retrospective study of parental feeding practices, we found that adults who had been pressured to eat as children or restricted in their calorie-rich snack foods were more likely than other adults to exhibit disordered eating and to have body image problems (Lev-Ari & Zohar, 2013b). In the current study, we therefore included parental feeding practices in our model of adult PE.

At the extreme, PE may be a symptom of avoidant/restrictive food intake disorder (ARFID, American Psychiatric Association, 2013). Thomas et al. (2017) suggest three neurobiological pathways to ARFID: Sensory sensitivity or sensory hyper-responsivity; low appetite, and fear conditioning. However, sensory sensitivity, low appetite and fear conditioning often co-occur in individuals diagnosed with ARFID (Reilly et al., 2019). Sensory hyper-responsiveness accompanies PE in normally developing children and adolescents, individuals with attention deficit disorder and people on the autistic spectrum (Smith et al., 2020). Since ARFID was introduced into the DSM-5 only in 2013 (American Psychiatric Association, 2013), there is limited research on the disorder and most published research focuses on children (Zickgraf et al., 2019). Adults who meet diagnostic criteria for ARFID may therefore be under-diagnosed. However, there are adults with picky eating in the community who would meet all the diagnostic criteria for ARFID. Zickgraf, Franklin & Rozin (2016) found in a community sample that included participants in a support group for self-professed adult PE, that 11.3% of the full sample met criteria for ARFID, as did nearly half of the participants ascertained from the picky eating support group. In this community sample most participants who reported eating less than 20 foods met criteria for ARFID. Therefore, the study sample of the current study may well have included undiagnosed adults with ARFID.

PE is related to responsivity to aversive sensory stimuli in both children and adults (Zohar, 2022). Correlational evidence for this finding is strong, and there is some experimental evidence (Madison & Stafford, 2023; Coulthard et al., 2022) for the responsiveness of adults with PE to sensory aspects of food presentation and appearance. This responsiveness to taste/smell/texture stimuli may be part of a more general sensory processing disorder (SPD). SPD is characterised by hyper and/or hypo- responsivity to various sensory stimuli–smell/taste, touch, position/movement, sound, and interoceptive stimuli (Galiana-Simal et al., 2020). Hyper-reponsivity to stimuli can be observed in young babies, both behaviorally and in terms of their altered brain activity (Adam-Darque et al., 2021). Usually diagnosed and treated by occupational therapists. there are clinical cut-off points for the disorder, in children as well as in adults (Bar-Shalita et al., 2009). For those with hyper-responsivity, the aversion to the stimulus is highly emotional, and disgust predominates (Harris et al., 2019; Brown et al., 2022). Moreover, individual differences in sensory sensitivity tend to be stable over development and may be part of a greater complex of traits that characterise “a highly sensitive person” (Aron, Aron & Jagiellowicz, 2012).

Disgust arose phylogenetically and has the selective advantage of protecting against pathogens and their toxic products (Toronchuk & Ellis, 2007), so it shapes food aversion. Even neonates exhibit disgust when presented with a bitter or sour food (Steiner, 1977). The basic emotion of disgust has physiological correlates of elevated heart rate and skin conductance (Stark et al., 2005). Disgust sensitivity is related not only to taste and smell, but also to auditory and tactile over-responsivity (Taylor et al., 2014). General and food disgust are in turn associated with eating disorders (Anderson et al., 2021) and play a role in PE (Brown et al., 2022). Brown et al. (2022) found that disgust sensitivity and propensity were related to selective eating and to symptoms of ARFID. There are theoretical reasons to believe that individual differences in disgust proneness and sensitivity are established early in development and remain stable throughout the life course. A short-term longitudinal study with adults supported this idea, showing that disgust proneness could be seen as a trait (Olatunji, Cox & Cole, 2020).

OCD is associated with PE. Schwarzlose et al. (2022) observed a strong association between PE and OCD symptoms for school-aged children. Many studies with adults have found a significant correlation between PE and symptoms of OCD (Wildes, Zucker & Marcus, 2012; Barnhart et al., 2021; Dial et al., 2021). Moreover, there is a strong association, and arguably a causal link, between disgust and OCD (Olatunji & Kim, 2024). It is possible to think of a system of sensory hyper-responsiveness, disgust, and food avoidance, all linked to a fear (conscious or not) of contamination, pathogens and disease. It is because of this putative system that we chose to include a measure of obsessive-compulsive symptoms in the current study, rather than symptoms of other pathologies than are elevated in individuals with PE.

The hypotheses of the current study were that adult PE scores would be positively and significantly correlated with aversive sensory hyper-responsivity, obsessive-compulsive symptoms, maternal feeding practices, and food and general disgust. We hypothesized that adults reporting having had PE in childhood would score higher than others on adult PE and the other study variables that correlate with adult PE. We hypothesized that it would be possible to build a model with a developmental rationale: Aversive sensory responsivity and disgust would lead to childhood PE and associated feeding practices, which in turn would lead to adult PE and obsessive-compulsive symptoms. We also hypothesized that we could identify a cut-off point on the APEQ, based on its relationship with categorical conditions such as childhood PE, probable OCD diagnosis, and probable SPD diagnosis. We also hypothesized that adult PE would seriously impact quality of life and therefore merit clinical attention.

Materials and Methods

Participants

Participants were sampled from Facebook© groups, including a national Israeli student group and groups of psychology students from six major Israeli universities, and snowballing in the community to include 772 participants with complete data (17.4%, n = 134 men). Their average age was 29.4 ± 8.5 years (range = 18–76). Mean length of education was 15.5 ± 2.5 years.

Measures

Propensity and sensitivity to disgust were measured by the Disgust Propensity and Sensitivity Scale-12 (DPSS-12; Olatunji et al., 2007). The DPSS-12 measures how often disgust is experienced (Disgust Propensity) and its emotional impact (Disgust Sensitivity). It was translated into Hebrew (with permission) for this study by two bi-lingual psychologists, using translation, independent back-translation, comparison and revision. Our data yielded reliability of α = 0.83 (total scale) and α = 0.82 for each of the two subscales.

Food disgust was measured by the Food Disgust Scale–short (FDS-S; Hartmann & Siegrist, 2018). The FDS-S has been validated against eating behavior (Ammann, Hartmann & Siegrist, 2018) and consists of eight items that assess the respondents’ emotional tendency to react to specific offensive, food-related stimuli. The FDS-S was translated into Hebrew (with permission) in the same way as the DPSS-12. In this study, its internal reliability was α = 0.77.

PE during adulthood was assessed by the Adult Picky Eating Questionnaire (APEQ; Ellis et al., 2017). The APEQ contains 16 items that assess PE behaviors and attitudes among adults. It comprises four subscales: Meal Presentation (MP), Food Variety (FV), Meal Disengagement (MD) and Taste Aversion (TA). The total score is calculated by averaging the responses to the 16 items. The APEQ was translated into Hebrew (with permission) in the same way as the DPSS-12 and the FDS-S. It yielded internal reliabilities of α = 0.88 (entire scale), α = 0.77 (MP), α = 0.83 (FV), α = 0.61 (MD) and α = 0.56 (TA).

Childhood PE was assessed by a single item: “In your childhood, were you a picky eater? (i.e., avoided certain foods and ate a narrow range of foods)?” Response options were yes/no.

Obsessive-compulsive behaviors and symptoms were assessed using the Revised Obsessive–Compulsive Inventory (OCI-R; Foa et al., 2002). The full OCI-R consists of 18 items, each of which describes a specific obsessive-compulsive behavior or symptom. The OCI-R includes six three-item subscales: Washing, obsessing, hoarding, ordering, checking, and neutralizing. Since DSM-5 criteria for OCD exclude hoarding, the three items about hoarding were not included. Wootton et al. (2015) validated the OCI-R against a clinical OCS diagnosis. A Hebrew version (Huppert et al., 2007) was used in this study, and the clinical cut-off point was set at 25. Internal reliability was α = 0.90 for the entire questionnaire and α = 0.70−0.90 for the subscales.

Retrospectively recalled parental child feeding behaviors were assessed using the Retrospective Child Feeding Questionnaire (RCFQ; Lev-Ari & Zohar, 2013a). We used three of the six RCFQ subscales: Restriction and monitoring of calorie-rich snack foods, as well as pressure to eat. The RCFQ has been validated in Hebrew against disordered eating, poor body image, and adult body mass index (BMI) (Lev-Ari & Zohar, 2013b). In this study internal reliabilities were α = 0.71 (Restriction), α = 0.67 (Pressure to eat) and α = 0.93 (Monitoring).

Aversive sensory responsiveness was evaluated using the Aversion subscale of the Sensory Responsiveness Questionnaire–Intensity Scale (SRQ-IS; Bar-Shalita et al., 2009). The SRQ-IS has been validated against a clinical diagnosis of sensory processing disorder (Bar-Shalita et al., 2009). In this study, internal reliability was α = 0.88.

Procedure

The research proposal was approved by the Ruppin Academic Center Ethics in Research Committee (#2021-175 S/cp). Written informed consent was given on the opening screen of the online study. Participants self-reported via Qualtrics XM. On the entry screen of the online survey, participants were told of the study goals, given contact addresses for consultation if they encountered any difficulties as a result of their participation, and asked to endorse their consent; if they did not they were directed out of the survey. Participants received no remuneration.

Data analyses

Hypotheses were specified prior to data collection. Associations were assessed using Pearson correlations (quantitative variables) and Chi-square (qualitative variables). ANOVA was used to assess group differences. A structural equation model (SEM) tested our hypothesized developmental model. We chose generally accepted values for a combined acceptance rule: Normed fit index (NFI) > 0.90 (Bentler & Bonett, 1980) and root mean square error of approximation (RMSEA) < 0.08 (Browne & Cudeck, 1992). SPSS 29 was used for all analyses with the exception of the SEM analysis which was conducted in AMOS 23. Statistical significance was set at p < 0.05. Data is accessable as a Supplemental File. IRB approval code 2021-175 S/cp.

Results

Hypothesis I: Adult PE scores will be positively and significantly correlated with aversive sensory responsivity, obsessive-compulsive symptoms, general and food disgust, and the retrospective maternal child feeding practices of pressure to eat, restriction and monitoring.

Correlation coefficients are presented in Table 1. The correlation between adult PE scores and aversive sensory responsivity, obsessive-compulsive symptoms, general and food disgust, and the retrospective maternal child feeding practices of pressure to eat, and restriction were all positive and significant. Maternal monitoring only correlated with the other feeding practices, and had a weak correlation with sensory hyper-reactivity. Hypothesis I was therefore largely supported.

Table 1 Inter-correlations between study variables (N = 771).

	SRQ-IS	OCI-R	DPSS-12	FDS-S	Restriction	Pressure	Monitoring	
Adult PE (APEQ)	0.36***	0.40***	0.56***	0.56***	0.09*	0.10**	0.00	
Sensory aversion (SRQ-IS)		0.42***	0.33***	0.27***	0.13***	−0.00	0.09*	
Obsessive-compulsive symptoms (OCI-R)			0.33***	0.30***	0.16***	0.16***	0.05	
General disgust (DPSS-12)				0.46***	0.11**	0.11***	0.06	
Food disgust (FDS-S)					0.05	0.02	0.03	
Maternal restriction (RCFQ)						0.15***	0.54***	
Maternal pressure to eat (RCFQ)							0.07*	
Mean	35.6	37.7	32.3	28.7	2.28	3.08	2.47	
SD	10.8	11.5	7.5	6.7	0.7	1.0	1.1	
Notes:

* p < 0.05.

** p < 0.01.

*** p < 0.001 (2-tailed).

PE, Picky eating; APEQ, Adult Picky Eating Questionnaire; SRQ-IS, Sensory Responsiveness Questionnaire; OCI-R, Obsessive-Compulsive Inventory-Revised; RCFQ, Retrospective Child Feeding Questionnaire.

Hypothesis II: Child PE, adult PE and variables associated with adult PE will be significantly associated.

We conducted a 2 × 7 one-way ANOVA with self-reported history of child PE (yes/no) as the independent variable and adult PE, aversive sensory responsiveness, obsessive-compulsive symptoms, general disgust, food disgust, maternal restriction and maternal pressure to eat as dependent variables. Maternal monitoring was excluded because its correlations with the other variables were insignificant. Group differences were significant: Overall (F(7,762) = 38.87, p = 0.001); adult PE (F(1,768) = 242.49, p = 0.001, η2 = 0.24); aversive sensory responsiveness (F(1,768) = 15.46, p = 0.001, η2 = 0.02); obsessive-compulsive symptoms (F(1,768) = 12.79, p = 0.001, η2 = 0.02); general disgust (F(1,768) = 96.29, p = 0.001, η2 = 0.11); food disgust (F(1,768) = 67.37, p = 0.001, η2 = 0.08); maternal restriction (F(1,768) = 8.16, p = 0.004, η2 = 0.01); and maternal pressure to eat (F(1,768) = 11.18, p = 0.001, η2 = 0.01). Means of the standardized variables for both groups are shown in Fig. 1. Hypothesis II was therefore supported.

Figure 1 Comparison of childhood picky eaters vs. non picky-eaters on standardized study variables.

Standardized study variables: Childhood picky eaters (N = 430) vs. others (N = 340) APEQ, Adult Picky Eating Questionnaire; SRQ-IS, Sensory Responsiveness Questionnaire; OCI-R, Obsessive Compulsive Inventory-Revised; DPSS-12, Disgust Propensity and Sensitivity Scale-12; FDS-short, Food Disgust Scale–short; RCFQ, Retrospective Child Feeding Questionnaire.

Hypothesis III: A structural equation model will tie early sensory-emotional responses (aversive sensory responsivity, general and food disgust) to early eating and feeding behaviors (maternal restriction of calorie-rich foods and pressure to eat; childhood PE; symptoms of OCD adult PE).

We built a SEM using AMOS 23 (Fig. 2). The Chi square goodness-of-fit index presented an excellent fit for the data, χ2(8) = 6.91, p = 0.55; NFI = 0.99; CFI = 1.00; RMSEA = 0.000.

Figure 2 A structural equation model of the development of adult picky eating.

SEM showing associations between the study variables (N = 711) For the path of general disgust–pressure to eat, p = 0.004; For the restriction–OCD symptoms, p = 0.02; For all other paths, p < 0.001. PE = picky eating.

Overall, the hypothesized model was confirmed. General disgust and food disgust directly influenced childhood PE. Maternal restriction was directly influenced by aversive sensory stimuli responsiveness. Maternal pressure to eat was directly influenced by general disgust. Food disgust directly influenced obsessive-compulsive symptoms and adult PE. Food disgust also influenced adult PE indirectly via childhood PE. Aversive sensory stimuli responsiveness, general and food disgust, maternal restriction and pressure, but not child PE directly influenced obsessive-compulsive symptoms.

Hypothesis IV: A data-based APEQ cut-off point for adult PE can be proposed, above which PE constitutes a problem that affects quality of life.

The APEQ includes 16 items rated on a Likert-like response scale of 1-5, and scores are means so that the potential range is 1-5. The actual range was 1–4.38, with mean = 2.23 + 0.68, mode = 2.13, and median = 2.12. The 75% cut-off point was 2.68 (mean plus half an SD). This score divided the sample into two groups, those below (N = 570) and above (N = 202) 2.68.

We tested the associations between this putative score with childhood PE, the clinical cut-off for sensory processing disorder (SRQ-IS), and the cut-off point for OCD (OCI-R); see Table 2.

Table 2 Associations of adult PE with childhood PE, aversive sensory modulation disorder, and OCD.

	Adult NON-PE	Adult PE	Chi-square; df = 1	Phi	
Childhood PE	4.4%	43%	147.2**	0.44**	
Sensory aversion	1.4%	9.9%	30.8**	0.20**	
OCD (OCI-R)	13.9%	37.1%	50.6**	0.26**	
Notes:

** p < 0.001

PE, Picky eating; % adult PE = Proportion of individuals who scored APEQ > 2.68 who reported childhood PE, and scored above the SRQ-IS cut-off and the OCI-R cutoff.

Results offer support for the upper 25% percentile or mean score of 2.68 as a cut-off for clinically significant adult PE. Those scoring above it report having had childhood PE, and most also scored above the suggested clinical cut-off for OCD on the OCI-R (Wootton et al., 2015) and had likely sensory processing disorder according to the SRQ-IS (Bar-Shalita et al., 2009).

Discussion

We examined the relationship between childhood PE, adult PE and associated variables: Maternal feeding practices, general disgust propensity, food disgust, obsessive-compulsive symptoms, and sensory responsivity to aversive stimuli. We found that retrospectively reported maternal restriction of calorie-rich snack foods and pressure to eat for the child were associated with adult PE, as were aversive sensory responsivity, OCD symptoms and disgust propensity.

Associations between adult PE, maternal pressure to eat, and disgust propensity have been previously observed (Ellis et al., 2016). Egolf, Siegrist & Hartmann, 2018 showed that food disgust was associated with PE, reluctance to try new foods, and food waste. The associations found in this study are therefore consistent with previous research.

Analysis of variance showed dramatic differences between adults who reported having PE as children and those who did not. Those with self-reported childhood PE scored higher in adult PE, aversive sensory stimuli responsivity, OCD symptoms, general disgust, food disgust, and retrospectively recalled maternal feeding practices of pressure to eat and restriction. Effect sizes ranged from small (maternal feeding practices) to large (adult PE). Childhood PE therefore appears to be associated with indices of maladaptive behavior and symptoms that extend into adulthood. This finding is also consistent with previous research, notably with findings from longitudinal studies that show long-term influence of childhood PE on adult eating behavior (Van Tine et al., 2017; Pereboom et al., 2023).

The structural equation model is consistent with developmental processes leading to adult PE. It supports the hypothesis that children who experience general and food disgust and are highly responsive to sensory stimuli that they perceive as aversive, are at risk for PE as adults. Maternal responses of pressuring them to eat, or restricting calorie-rich snack foods are usually counterproductive. Far from “curing” childhood PE, they may even exacerbate it (Zohar, Lev-Ari & Bachner-Melman, 2021). These maternal feeding practices and child aversive sensory responsiveness and disgust propensity may contribute to OCD symptoms, and all these factors may contribute to adult PE. A possible developmental process is that children who experience disgust and possible fear of food develop OCD symptoms in a maladaptive attempt to control their environment and improve its predictability (Zohar & Felz, 2001). However, it is important to emphasize that a structural equation model is basically a correlational analysis that requires longitudinal and experimental support for verification.

This study identified a putative clinical cut-off for adult PE, based on the APEQ. Scoring in the top 25% of respondents for adult PE was significantly associated with having a probable OCD diagnosis, reporting a history of childhood PE, and being at increased risk for sensory processing disorder. The effect size for these associations ranged between small for sensory processing disorder to medium for OCD and childhood PE. Caution is called for in relation to this finding, since this cut-off point may be specific to this sample or culture. Further validation studies with more diverse populations are therefore required to determine a clinical cut-off point for adult PE using the APEQ. Existing data sets (e.g., Barnhart et al., 2021; Fox et al., 2023) could be re-accessed and cut-off scores determined by pooling correlates.

The results of this study could be integrated as follows: Individuals who are prone to feeling disgust, and who respond with negative emotion and disgust to sensory stimuli, may tend to be more restrictive and selective in their eating, from childhood through to adulthood. Well-meant parental feeding practices that strive to extend the variety and quantity of child food consumption tend to backfire. Since fear of contaminants and disgust is closely associated with obsessive-compulsive tendencies, adults with the profile described above tend to display obsessive-compulsive symptoms and PE. Sensitive/neophobic children (Zickgraf et al., 2019), whether or not their PE is a symptom of ARFID, are channeled more easily (though inadvertently) than other children into a developmental path that progress into adult OCD, adult sensory processing disorder, and adult PE and/or ARFID. It seems that as in many other eventualities, early intervention would be effective, helping families deal with their children’s PE and sensory hyper-responsivity by avoiding accommodation to PE (Shimshoni, Silverman & Lebowitz, 2020), and to sensory hyper-responsivity (Ben-Sasson, Podoly & Lebowitz, 2023; Walbam, 2023) and by adopting more helpful approaches to parental feeding (Zohar, 2022).

The results of this study should be considered in the context of the study limitations. The participants were recruited on a volunteer basis and the sample is neither random nor representative. As in all volunteer samples, women outnumber men. All data was self-reported with all the attendant biases of self-report. Moreover, data was collected at a single time-point so that it is not possible to establish precedence. Childhood PE was assessed retrospectively by a single item, and parental feeding practices were also assessed retrospectively. Inferences about developmental processes should therefore be considered as testable hypotheses. Moreover, food avoidance and selectivity is a feature of the eating behavior of individuals on the autistic spectrum (Dovey et al., 2019), as well as individuals with ADHD (Kaisari, Dourish & Higgs, 2017), and inherently of individuals with ARFID (American Psychiatric Association, 2013, p. 334). We did not include measures of these pathologies in the current study and these might have explained considerable variance in adult PE in this study.

Future research for determining what perpetuates PE from childhood to adulthood might be tested by large-scale, longitudinal studies. Another more feasible approach would be to use ecological momentary assessment, which could test the influence of momentary rise of food disgust to avoiding certain foods, or the momentary contribution of obsessive-compulsive symptomology to food avoidance. This approach has been suggested in the guidelines to the maintenance of all eating disorders, trans-diagnostically (Levinson et al., 2024).

Conclusions

To sum up, adult picky eating is associated with aversive sensory responsivity, obsessive-compulsive symptoms, general and food disgust, and a childhood history of PE. In addition, having been pressured to eat as a child or limited in the consumption of calorie-rich foods also appears to bear some connection to PE in adulthood. Longitudinal research is required to determine which of these variables are in fact risk factors for adult PE. However, for a sizable minority of adults, it seems that PE is of clinical significance and is therefore worthy of further study so that effective interventions for adult PE can be developed and implemented.

Supplemental Information

Supplemental Information 1 Notes on dataset.

Supplemental Information 2 Codebook.

Supplemental Information 3 Raw data.

Supplemental Information 4 Categories of response for the questionnaires used.

Additional Information and Declarations

Competing Interests

Ada H. Zohar is an academic editor for PeerJ.

Author Contributions

Ada H. Zohar conceived and designed the experiments, analyzed the data, prepared figures and/or tables, authored or reviewed drafts of the article, and approved the final draft.

Dorin Barhum Shapira conceived and designed the experiments, performed the experiments, prepared figures and/or tables, and approved the final draft.

Lilac Lev-Ari conceived and designed the experiments, analyzed the data, prepared figures and/or tables, and approved the final draft.

Rachel Bachner-Melman conceived and designed the experiments, authored or reviewed drafts of the article, and approved the final draft.

Human Ethics

The following information was supplied relating to ethical approvals (i.e., approving body and any reference numbers):

Ruppin Academic Center Ethical Committee (#2021-175 S/cp).

Data Availability

The following information was supplied regarding data availability:

The raw data is available in the Supplemental File.

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
