# Peer review of "Adult picky eating and associations with childhood picky eating, maternal feeding, aversive sensory responsiveness, disgust and obsessive-compulsive symptoms"

_PeerJ, doi:10.7717/peerj.19444_

## Round 0.1 · original submission · Major Revisions

Dear Authors,

Thank you for submitting your paper to this Journal. Your work has merit, however the reviewers have recommended several revisions to the text before it can be considered for publication.

·

Basic reporting

I have several concerns related to literature references/sufficient context being provided. Other areas are accpetable in the current draft.

This study includes interesting hypotheses and a novel combination of variables, but overall I felt that the organization of the paper made it a confusing read. The introduction overlooked a lot of literature on adult PE (e.g., its relationship with ARFID, previous findings on the relationships between and among study variables relevant to the hypotheses) and failed to justify many of the hypotheses. The literature review was mostly limited to one or two citations for the existence of a relationship between PE and some, though not all, of the variables that ended up in the hypotheses, but did not usually contextualize or justify the authors’ specific hypotheses. Nothing is said about the relationship between PE and maternal feeding practices, sensory aversiveness and disgust, or PE and OCD until the hypotheses themselves are introduced.


The authors begin by stating that adult PE is not a disorder, which is true, but they do not mention that PE is one of the three eating restrictions that can lead to a diagnosis of ARFID. The ARFID field currently recognizes three manifestations of ARFID (that can overlap/co-occur), of which one is picky eating, referred to in the ARFID context as “sensory sensitivity” (Thomas et al., 2017) or “selective/neophobic eating” (which is my preferred terminology!). Given that ARFID researchers treat PE as a spectrum with ARFID at the extreme end, a discussion of ARFID is warranted in the introduction, and the authors should consier addressing how their proposed model predicting adult PE might relate to ARFID risk, psychopathology, and treatment. I would argue that studying PE without considering ARFID is akin to studying contamination fear without considering OCD.

Paragraph beginning line 106 – disgust sensitivity has been found to mediate the relationship between anxiety and PE, and is strongly related to both PE and ARFID symptoms in adults (Harris et al., 2019). Food disgust sensitivity is more strongly related to PE than general disgust propensity, and general disgust sensitivity is not related to PE (Brown et al., 2022). While being succinct, the introduction should include the literature most relevant to the current study.
Brown, T. A., Menzel, J. E., Reilly, E. E., Luo, T., & Zickgraf, H. (2022). Exploring the role of disgust sensitivity and propensity in selective eating. Appetite, 174, 106018.
Harris, A. A., Romer, A. L., Hanna, E. K., Keeling, L. A., LaBar, K. S., Sinnott‐Armstrong, W., ... & Zucker, N. L. (2019). The central role of disgust in disorders of food avoidance. International Journal of Eating Disorders, 52(5), 543-553.
The hypothesis about parent feeding practices isn’t set up by the introduction. Suggest a quick summary of the literature on childhood PE and parent feeding practices. There have been several retrospective studies on this using adult report, in addition to studies in children.
108-109 – can the authors define aversive sensory responsiveness here, and elaborate a bit on their moderation hypothesis?

The authors introduce BMI in the introduction, but don’t have hypotheses about how it relates to PE. Given relationships between coercive parent feeding practices (hypotheses I and IV) and higher weight in some studies, I expected the authors to test whether these factors might moderate the relationship between childhood PE and adult BMI. The amount of space dedicated to BMI in the introduction, especially relative to some of the other predictors that are included in the hypotheses, doesn’t seem appropriate given that BMI isn’t a study variable.

It would also be interesting to explore how coercive parent feeding practices moderate the relationship between childhood PE and adult PE. Do these parent practices actually “work” in that they result in less persistence of PE, or is the relationship between childhood and adult PE strengthened by elevated coercive parenting around food? This idea is raised in the discussion (lines 291-295) but it seems like the authors could test this hypothesis with the data they have.

In general, the introduction sets the reader up to expect the study to be about factors that explain concurrence between childhood and adult PE—what factors distinguish transitory, developmentally appropriate PE from PE that persists into adulthood?

Experimental design

I have a few questions/concerns about the overall aims and specific analyses:

Can the authors provide justification for treating the APEQ as unidimensional in their aim regarding identifying cut-offs? The scale has factors and to my knowledge no hierarchical or unidimensional solutions have been published. In this large a dataset, there should be enough power to explore whether there is a single latent first or second order factor!

Lines 108-109 seem to imply mediation more than moderation (sensory responsiveness “precedes” disgust developmentally). In the discussion, lines 300-303, the relationship is again described in a way that sounds more like mediation (higher sensory responsiveness leads to higher food disgust (via more opportunities for conditioning), and therefore higher picky eating) than moderation (e.g., for people higher in disgust sensitivity, there is a stronger relationship between sensory aversiveness and PE. This could suggest that people who are more prone to food disgust tend to experience their heightened awareness of food sensations as disgusting, and therefore engage in PE behavior, whereas for those who are less prone to being disgusted by food, sensory acuity is less strongly linked to PE behavior).

Minor point: The wording of hypothesis IV is confusing – are symptoms of OCD and adult PE really considered “early eating and feeding behaviors”? Based on how the SEM is structured, I think they should appear outside the parenthesis as a separate item on the list?

Results – Hypothesis II, Hypothesis III, Table 2: please report effect sizes for all analyses. For the moderation analysis, please report effect sizes for the moderation effect and main effects. In the discussion you should address how strong the moderation effect is compared to the main effect of food disgust.

Overall, I have concerns with how the SEM in Hypothesis IV is described. All of the variables other than childhood PE and recalled maternal feeding practices are measures of current traits, experiences, and behavior. The authors don’t really make the case that disgust sensitivity, sensory sensitivity, and food disgust are traits that consistently or by definition develop in early childhood and remain relatively stable over development. This is probably true, but not a matter of fact that doesn’t need to be justified and appropriately cited. Even then, the fact that these measures are all worded to ask about the respondent’s CURRENT disgust sensitivity/propensity, food disgust, and sensory responsivity is a limitation to the mediation model that implies temporal ordering. Maybe it would be more appropriate to make mechanistic hypotheses about what maintains picky eating rather than risk/developmental hypotheses about what causes it? This could set up testable EMA hypotheses (e.g., momentary picky eating behaviors are strongly influenced by the tendency to be disgusted by food or by the momentary experience of food disgust). These are more feasible to test than longitudinal hypotheses that would require the variables of interest to be available from cohort studies.
Why is there no path between food disgust and maternal feeding practices? It seems like food-specific disgust is theoretically more relevant to feeding practices than general disgust. Was there an a-priori choice not to include this path, or was it dropped because it wasn’t significant?

Validity of the findings

The discussion failed to contextualize the findings in the broader literature or connect them to specific implications, such as the psychopathology of PE, the treatment of ARFID caused by PE or of subclinical PE, how to prevent childhood PE from progressing into adult PE, or the potential mechanisms of the relationships between PE and other psychopathology that was included in the study (OCD, sensory processing disorder) and other psychopathology that was not included but might be relevant (e.g., anxiety, whose relationship with PE is mediated by disgust in several studies) or might reflect a third variable in the relationships among PE/OCD/SPD observed in the study (i.e., autism).

295-296: OCD is usually understood to be a neurological condition that is present early in development. Therefore, rather than disgust propensity and food fears leading to the development of OCD, the reverse might be true, or all three might be attributable to shared risk/vulnerability factors like neuroticism, fear conditionability, and cognitive rigidity.

Reviewer 2 ·

Basic reporting

The manuscript is written in clear, professional English. However, some minor instances of awkward phrasing could be refined.
In the Abstract and introduction the term “Adult community volunteers” is vague, what exactly is meant by this wording? Do you simply mean ‘participants’? If so I would stick with the common terminology.
The last sentence of the abstract should read Extreme adult PE is closely related “to” childhood PE. This appears correct in the full-text version but not on the abstract page.
Line 78: I would add - picky eating (PE) is not “defined as” a disorder.
Line 82 “binge eating” is more specific than “bingeing”.
Line 85 – amend “selection in food choices” to “food selectivity”, which is a more commonly used term.
Line 95- remove ‘the’ before age.
Line 133: Spelling error “include”
I would avoid the terminology “picky eaters” and use person-first language throughout the manuscript as it is less stigmatizing (i.e., adults with picky eating).

Experimental design

Originality and Scope:
The research question is well-defined, addressing a clear knowledge gap regarding the developmental factors influencing adult PE.
Methodology:
Participant recruitment is adequately described, but the non-random sampling method (via social media and convenience sampling) limits generalisability.
The statistical analyses (correlations, ANOVA, structural equation modelling) are rigorous and appropriate for the study aims. However, the justification for excluding maternal monitoring from certain analyses is somewhat cursory and could be expanded.
Section 204-215: What corrections were applied for multiple testing?
Section 220-229: Which model was used in the macro Process? You should specify the model number for transparency.
A major limitation is the large proportion of females in the study.
The omission of data on autism spectrum disorder (ASD) or other forms of neurodivergence represents a significant limitation in the study, particularly given the known associations between sensory sensitivities and these conditions. By not including data on neurodivergent diagnoses, the study misses an opportunity to identify specific interventions or support strategies for individuals who may face compounding challenges due to their neurodivergence and PE.

Validity of the findings

Data Quality:
The data appears robust and sufficiently detailed to support the conclusions. However, the reliance on self-reported measures introduces the risk of bias. The statistical methods are appropriate, and the results are presented with sufficient detail. Fit indices for the structural equation model indicate a strong model.
Interpretation of Results:
Section 251-258: I don’t think that the data analysis can appropriately support hypothesis 5 and I would suggest potentially excluding this hypothesis from the manuscript given that the measure is not validated against any existing measures for picky eating. Without further validation and confirmatory analysis, including clinical diagnosis of a disorder such as ARFID or EDNOS, I think that it is too strong to suggest that a cut-off point can be proposed. Additionally, further validation studies in diverse populations are necessary before it can be widely applied.
Conclusions are well-supported by the data, but the authors should more explicitly acknowledge the limitations of cross-sectional data in establishing causality.

Additional comments

I would expect in the first paragraph an explanation of Avoidant Restrictive Food Intake Disorder which is now recognized. This paper needs to clarify what picky eating means in relation to ARFID which is the extreme version of picky eating and already proposed as a clinical cut-off for picky eating.
Lines 98-101 is a very broad generalisation and does not address the role of other factors that could influence associations between PE and BMI. Food preferences among individuals with higher weight are highly heterogeneous and influenced by a variety of factors beyond simple preference, such as socioeconomic status, access to food, cultural factors, and metabolic differences. The statement risks reinforcing stereotypes about eating behaviours in high-weight individuals. You should clarify and expand on the mechanisms linking childhood PE to adulthood outcomes, incorporating a more balanced view of individuals across different weight categories and contexts.
The use of one item to measure retrospective picky eating is a major limitation to the work and needs to be addressed. Furthermore, there is no indication of child age, since eating behaviors can be transient and picky eating in children aged 2-7 is very common but tends to peak at 7 years.
Retrospective parental feeding practice measure is also a major limitation to the study and should be addressed. Also there needs to be more justification as to why only three ‘coercive’ feeding practices were assessed and not more structured or autonomy support practices.
The limitations section needs to be majorly revised to include the above remarks, (i.e., lack of ARFID measures, lack of neurodivergence data, predominantly female sample, single item measures, recall bias of childhood measures, cultural specificity of findings).

·

Basic reporting

Overall, this paper is well written. The authors demonstrate that childhood picky eating is related to adult picky eating. The authors also demonstrate the relationship between picky eating and OCD symptoms, sensory modulation disorder, disgust, and maternal childhood feeding practices. Picky eating in adults is often overlooked in the literature, and I applaud the authors for examining this group that is less researched.

Abstract
1. Please spell out OCD in the abstract at the first mention

Introduction
1. Please put the hypothesis in paragraph format.
2. Relatedly, hypotheses 4 and 5 don't feel like hypotheses but more so data analytics please revise

Methods
1. Can authors make a clear data analytic plan selection
2. Why did the authors use two different statistical software?
3. Can authors make it explicit which software was used for which analyses?

Results
1. On line 198 can authors explicitly list out risk factors.
2. For hypothesis 5 lines 253-254 that first sentence feels like it does not belong there but rather in the methods section when describing the measure

Discussion
1. Please list out strengths/limitations

Experimental design

No comments

Validity of the findings

no comments

Additional comments

not comments

---

## Round 0.2 · Minor Revisions

The authors have addressed most of the reviewers' concerns. Some minor revisions have been suggested by one of the reviewers.

·

Basic reporting

I really appreciate the authors' engagement with reviewer concerns in this revision. I think the introduction does a great job of contextualizing the hypotheses and I appreciate the addition of ARFID.

I just have a remaining quibble with the treatment of ARFID in the introduction. Selective eating, or rejection of foods based on their sensory properties, is one of three presentations that each separately can cause someone to meet impairment criteria for ARIFD (Thomas et al., 2017), but which often co-occur in people diagnosed with ARFID (e.g., Reilly et al., 2019, Norris et al., 2018, Cooney et al 2018). The paper of mine that you cited (Zickgraf, Burton Murray et al, 2019) was not a pediatric eating disorder clinic sample, it was an outpatient anxiety and OCD clinic. In a separate chart review from a pediatric eating disorder clinic (Zickgraf, Lane-Loney et al., 2019), there were very few patients with picky eating as their only presentation of ARFID, a minority with co-occurring picky eating and appetite impairment, and around half had fear-ARFID as their only presentation. Because the setting for Zickgraf Burton Murray et al., 2019 was an anxiety clinic, we actually excluded patients who met criteria for ARFID because of fear of aversive consequences of eating from the chart review because we suspected that anxiety/OCD clinicians might be less likely to diagnose this presentation as ARFID, instead conceptualizing it just as an anxiety disorder. Taken together I don't think either of those papers can support the statement that "children meeting ARFID criteria can be best described as selective/neophobic eaters."

On a related note, I don't think the APA or Thomas et al., 2017 citation support the statement "Children and adults with ARFID are, by definition, highly selective and avoidant in what they eat and their “avoidance is based on the sensory characteristics of the food” (APA, 2013, p334)." The quoted text is describing one of the three presentations, not ARFID in general.

"Brown et al. (2022) found that disgust sensitivity but not propensity was related to selective eating and to symptoms of ARFID."
- Both were actually related to selectivity, but the correlation with propensity was stronger than that with sensitivity. In a regression with the food disgust scale in the model, sensitivity was actually weakly inversely related to PE and propensity was independently predictive of PE.

Experimental design

The authors were responsive to reviewer concerns! I still recommend reporting effect sizes with all statistical findings, not just in tables (e.g., the ANOVA results for hypothesis ii in the text)

Validity of the findings

In the discussion: "Children who experience disgust and possible fear of food might develop OCD symptoms in a maladaptive attempt to control their environment and improve its predictability (Zohar and Felz, 2001)."

I'm not sure I agree with this interpretation of the findings given that there was no path from PE to disgust.

I really liked the summary of the findings given starting in line 415, which treated PE and OCD as disorders with shared risk factors/developmental trajectories, rather than one (PE) causing the other (OCD).

Reviewer 2 ·

Basic reporting

No comment

Experimental design

No comment

Validity of the findings

No comment

Additional comments

Thank you for the opportunity to review the revised manuscript. I am happy to confirm that the authors have addressed my feedback thoroughly and thoughtfully. The amendments made have improved the clarity and rigor of the manuscript, and I am satisfied with the responses and revisions provided.

---

## Round 0.3 · accepted · Accept

The reviewers have recommended accepting the paper.

·

Basic reporting

NA

Experimental design

NA

Validity of the findings

NA

Additional comments

Thanks for your engagement during the review process! I'm looking forward to citing your work!